# Ultrasound Elastography—Cornerstone of Non-Invasive Metabolic Dysfunction-Associated Fatty Liver Disease Assessment

**DOI:** 10.3390/medicina57060516

**Published:** 2021-05-21

**Authors:** Andrej Hari

**Affiliations:** Department of Gastroenterology, General Hospital Celje, 3000 Celje, Slovenia; andrej.hari@sb-celje.si

**Keywords:** liver steatosis, elastography, liver fibrosis, portal hypertension, non-invasive assessment

## Abstract

Metabolic dysfunction-associated fatty liver disease has become the most common chronic liver disease as well as the most common cause for liver transplantation. With its different methods types, elastography of the liver can be used for non-invasive evaluation of the liver fibrosis and steatosis degree. The article focuses on the description, use, advantages, and limitations of the currently known elastographic techniques. It proposes a simple risk assessment algorithm for the liver fibrosis progress evaluation. The following is an overview of the use of liver and spleen elastography in the detection of clinically relevant portal hypertension. It concludes with research and technological possibilities that could be important to the field in the upcoming years.

## 1. Introduction

Liver steatosis is a pathological fat accumulation in hepatocytes of the liver. It is caused by a variety of causes, of which by far the most important and most common is the one resulting from the metabolic system dysfunction [1]. The metabolic syndrome pandemic has elevated cirrhosis caused by non-alcoholic steatohepatitis (NASH) to the forefront cause of liver transplantation [1]. In recent years, we have consequently recorded important milestones when approaching this medical entity, especially those intended for an early and accurate disease evaluation. The first is an attempt to define more precise diagnostic criteria for a condition that has so far been based on the exclusion of other chronic liver diseases and known as non-alcoholic fatty liver disease (NAFLD). A proposal by Eslam M et al. was made to replace the NAFLD term with the metabolic dysfunction-associated fatty liver disease (MAFLD), which allows a more precise definition and allows a simultaneous presence of several etiologically different chronic liver diseases (Figure 1) [1]. Expert opinions regarding the naming and definition appropriateness are, of course, diverse in the early response stages. In a sense of an article better recognition, we use the term MAFLD.

Another important achievement coincides with the disease’s prevalence. MAFLD is a very common condition in the adult population, but on the other hand does not show frequent progression to liver cirrhosis (1–7% long-term chance). Diagnostic evaluation of such a condition has become both a logistical and a financial burden to the health system [2]. Adequate MAFLD risk stratification and management requires assessment and quantification of liver parenchymal fat, inflammation, and fibrosis, which is most reliably performed by histological examination of liver tissue. A modified Brunt or SAF (Steatosis, inflammation Activity, Fibrosis) score is most common used for histological assessment of changes. Criteria of inflammatory cell infiltration, hepatocyte necrosis and ballooning, and perivenular fibrosis in addition to steatosis are important for histological assessment of steatohepatitis (NASH), while MAFLD shows variable degrees of steatosis, but neither necroinflammation nor ballooning of hepatocytes. The presence and degree of inflammatory activity and NASH can currently only be assessed by histological examination and is recognized in lobules and/or portal tracts with a predominantly centrilobular distribution [3]. Due to the known deficits of liver biopsy (invasiveness, sampling error, static assessment, need for an experienced pathologist, interobserver variability, patient anxiety), diagnostic methods of non-invasive liver assessment are coming to the fore. In the group of blood biomarkers/serum scores, MRI/CT imaging and ultrasound, especially elastographic methods, this article will focus on the last diagnostic methods mentioned—ultrasound methods.

## 2. Ultrasonography and MAFLD Evaluation

The cornerstone of modern investigative methods in this field is liver elastography. The article briefly summarizes some other methods that may be part of advanced ultrasound features.

The basic principle of elastography is that an acoustic impulse passes through the liver tissue and acts as a wavefront that causes minimal displacement of the tissue. This leads to the formation of shear waves in the liver tissue which spread faster in a stiff medium (i.e., cirrhosis) [4].

Figure 2 shows the most common used elastographic methods. Unless otherwise stated, the term elastography or elastographic examination in this article refers to the most common used method—transient elastography (TE; Fibroscan©).

TE is a unidimensional method in which a simultaneous ultrasound (US) representation of the investigated tissue is obtained as a simple M-mode picture (Figure 2). Other methods have integrated the elastography module into conventional abdominal US probes. This enables a morphological analysis of the organ in the grayscale with a superimposed measuring box in which the liver stiffness is measured. In point shear wave elastography (pSWE), the measuring box is small in size and there is no visible elastogram (Figure 2). If the measuring box is larger and has a visible elastogram (every point in the elastogram is color coded and represents different shear wave speeds), the method is called two-dimensional shear wave elastography (2D-SWE; Figure 2) [4]. The common advantage of all three modalities is that they are non-invasive, point of care and able to evaluate a greater volume of liver parenchyma with lesser variability and more quantitative specimen evaluation than liver biopsy which reduces sampling error in heterogeneously distributed liver disease processes [3].

The proposed reliable elastographic measurement requires a minimum of 10 valid (5 in 2D-SWE technique) measurements, in which case >60% of all measurements should be valid and the ratio of the median valid liver stiffness measurement (LSM) to interquartile range value (IQR) should be less than or equal to 0.3 [5]. Results indicating IQR/LSM > 30% in conjunction with LSM ≥ 7.1 kPa are particularly unreliable. IQR/LSM ≤ 0.1 with any LSM value indicates very reliable measures. IQR/LSM > 0.1 and ≤0.3 with any LSM value or IQR/LSM > 0.3 with LSM < 7.1 kPa are considered to indicate reliable measures [5]. LSM confounders that are independent of fibrosis include inflammation, cholestasis, food intake, and portal vein thrombosis. Elevation in ALT (>3 times above the normal value), alcohol excess, use of diuretics, infiltrative liver diseases, or other comorbid conditions that lead to hepatic congestion (right heart failure) will also significantly elevate LSM value [6,7]. TE is studied much more extensively than pSWE or 2D-SWE, but they all appear to have similar accuracy [3]. Intraobserver agreement for TE is excellent, although it is lower with lesser degrees of fibrosis, increasing steatosis, and BMI [8]. Around 15% of results may be unreliable, and failure to obtain any LSM occurs in ∼3% of patients, mostly due to obesity or operator inexperience [6,7]. pSWE has a very low scan failure rate (0–1%). However, it is unreliable in 16–24% of subjects and has a learning curve, with intraobserver agreement increasing after 100 examinations. 2D-SWE does not have validated reliability criteria and thus invalid scans are typically not reported, although it has a failure rate of 1–13%, being higher in patients with MAFLD. 2D-SWE also require some degree of radiological expertise compared to TE, with greater intraobserver variability noted in less experienced operators [9]. It should be noted that there is a lack of quality criteria, uniformity in commercial system design, variability in shear wave frequency, sampling rates, and other technical parameters that limit the comparison of LSM across manufacturer systems [6,7,10]. Ultrasound elastography embedded in conventional scanners usually allows the choice of where to place the region of interest (ROI) stiffness box and whether to confirm or exclude each single measurement when determining the final value. Thus, the operator has a greater potential to influence the final findings than with TE where these choices are not available. Finally, aforementioned factors affecting the final results have not yet been fully reported for many of the latest equipment, as opposed to the methods available for more than five years [10]. However, a study on phantoms recently showed that there is a significant difference in LSM estimates among systems, but no statistically significant differences were found among observers using the same system, and they also reported very good agreement between systems [11]. Due to the high dependency of the proposed cut off values on the specific elastographic technique, it is important to notice that the following is a revision of mainly TE derived cut off indices. Head to head study comparisons have proven that when determining the applicable clinical values, the cut off indices would be usually of slightly higher values in the TE area when compared to the pSWE and 2D-SWE data. Study interpretation would suggest that the impact of the size of the evaluated liver tissue area might explain some of the differences between the methods [3,4,6,8].

As can be seen from the above mentioned references, the field of ultrasound elastography is apparently divided into various investigative methods, the advantages, disadvantages and differences of which must be known in detail by every clinician that would like to become expert in the field. In the following subsections, the article deals with the clinical applicability of elastography namely its use when assessing the degree of liver fibrosis, liver steatosis and the presence of clinically important portal hypertension. Due to the high dependence of clinical applicability and reliability of elastographic methods on reported statistical indices (AUROC studies, positive and negative predictive value of the result—PPV and NPV), the following terms are used to facilitate comprehensibility of the article, replacing numerical reporting: <90% PPV or NPV—moderate accuracy; >90% PPV or NPV—high accuracy.

## 3. Elastographic Assessment of Liver Fibrosis

For easier comparison, elastographic results are often equated with histological classes of liver fibrosis rate used in the Metavir score (F0–4; F0 and 1—absence of significant liver fibrosis; F3 and 4—advanced liver fibrosis). A value of 4 or 5 kPa is usually proposed to define the absence of any liver fibrosis (F0/1). As in the other fields, it is important to assess the absence of a clinically significant liver fibrosis (<F2) in a MAFLD patient and to define the presence of an advanced stage of liver fibrosis that clinically classifies patients to the compensated advanced chronic liver disease (cACLD; ≥F3). According to the performed meta-analyses of published studies, at fixed sensitivity, a cut-off of 6.5 kPa is proposed to exclude the presence of significant fibrosis and a cut-off of 12.1 kPa is required to have a high specificity when we want to define cACLD [5].

When reviewing the study literature, we can observe a considerable heterogeneity of the reported cut-off values. A meta-analysis by Cai C et al. suggested 6.3 kPa, 8.2 kPa, 13.4 kPa, and 14.2 kPa as cut-off values for MAFLD patients with fibrosis grades ≥F1, ≥F2, ≥F3, and F4 [12]. A study by Eddowes et al. points out that the cut-off for liver cirrhosis (F4) is markedly higher at 20.9 kPa when the specificity is set at 90%. In the same study, cut-off values for F2, F3, and F4 as >8.2 kPa, >9.7 kPa, and >13.6 kPa were defined. These cut-off values have good sensitivity and specificity with moderate PPV for F2 and high NPV for F4 [13]. An important study in this area made a comparison between the histological liver fibrosis assessment and the TE results in several thousand patients from ten tertiary centers. It estimates that the prediciton value of each assessment can be improved by setting two basic cut-off values. They propose a dual cut-off of <7 kPa and >12 kPa to optimize the overall predictive performance of TE when assessing the presence or absence of cACLD and with which they were able to correctly classify more than 80% of patients with chronic liver disease. In the gray zone, a formula taking the patient’s age, male gender, ALT/AST/yGT values, platelet counts, and the presence of type 2 diabetes into account was proposed [14]. Important study observations show that the reliability of the elastographic measurement increases if two consecutive measurements are performed in a time window of 3–6 months. As reported by the group by Chuah et al. one-third of patients with LSMs > 12 kPa at first determination had normal values at a second measurement performed 4–6 months later. One-third of patients with high LSM may have normal results on repeated examination, whereas a persistent increase in liver stiffness increased PPV to 65%. By repeating examination in cases with high LSM, one may spare patients from unnecessary liver biopsy. It is particularly interesting that none of the patients with a body mass index (BMI) < 30 kg/m^2^ had a falsely high result. In this way, less than two percent of the clinically significant fibrosis was missed during the pairwise measurement and it was determined that a repeat LSM is not necessary when baseline LSM is ≥20 kPa [15].

When dealing with a MAFLD patient, elastography is more reliable for diagnosing cirrhosis (F4) with the correct classification in 80–98% of cases than it is for advanced fibrosis (≥F3). In other words, almost 30% of cases diagnosed as cirrhosis are actually false positive. However, the method is very reliable for ruling out cirrhosis, with a very low number of false-negative results (<6%) following stiffness measurements with values below the established threshold [4].

An important area that causes difficulties in obtaining a reliable result is obesity, which is very common in the group of MAFLD patients. To combat this problem, TE launched a two-probe system (M and XL) a couple of years ago where the system automatically proposes an XL probe in case it assesses the significant impact of skin and subcutaneous thickness on the result [4]. Recently, it was shown that 2D-SWE has a good accuracy, success rate, and diagnostic performances in MAFLD patients with obesity. A two-step strategy using TE followed by 2D-SWE was reported to be even more accurate in cACLD detection. Multi-step strategies using 2D-SWE may also significantly reduce the need for liver biopsy [6,11]. Individual authors cite a significantly extended measurement time as a disadvantage of such evaluation [11].

The result and thus the proposed threshold values for TE are significantly different when using an M or XL probe (lower when using an XL probe). An important influence when obtaining a reliable measurement by TE in the field of MAFLD is the concomitant presence of liver steatosis. Reports in the literature regarding the effect of steatosis on the elastographic result are somehow contradictory. Thus, certain authors advise that a special consideration should be given to the importance of steatosis severity in contributing to liver stiffness in this condition [15,16]. The reported effect of steatosis on pSWE seems to be the lowest between the three mentioned techniques [16]. On the other hand, an extensive recent study in this area demonstrates that only fibrosis stage, and not probe type or any other histological parameters, including liver steatosis, influence LSM values [13].

A simplified flowchart summarises the discussed topic and is proposed for the everyday clinical use (Figure 3). Note the important disadvantage of the chart—evaluation of the inflammation (NASH) presence and the inflammatory activity degree. Inclusion of proposed formulas reported by Papatheodoridi et al. [14] is recommended.

## 4. Elastographic Assessment of Liver Steatosis

A standard assessment for the presence of liver steatosis is performed histologically. Histological grades (S0–S3) follow each other according to the estimated percentage (%) of steatosis-affected hepatocytes in the histological sample. Absence of liver steatosis (S0) is defined as <5% hepatocyte involvement (<10% for study purposes), followed by grades S1 (>5% and <33%), S2 (>33% and <66%), and S3 (>66%). In clinical practice, major milestones in the liver steatosis course are its absence (<S1) and the presence of a clinically significant liver steatosis grade (≥S2), which has a potentially greater impact on the course of the disease and metabolic risk of the patient [1].

MRI Proton Density Fat Fraction (MRI-PDFF) is regarded as the most definitive non-invasive imaging tool for qualitative and quantitative evaluation of liver steatosis with both high specificity and sensitivity for detecting any grade of histologically confirmed steatosis. Due to its non-invasiveness, it is used as a comparative standard in studies, but due to its poorer accessibility and price, it is usually replaced by other methods in everyday clinical practice [3,17].

Fatty liver has higher US echogenicity than renal cortex and is detectable by standard US modality when more than 20% of hepatocytes contain histologically visible fat droplets, with moderate sensitivity and specificity [3,18].

Strictly technically speaking, the following is not an overview of the elastographic tecniques since the presented data is a computer calculation of the signal that evaluates liver tissue either by TE or by US B mode signal. In a much simplistic explanation, all of the mentioned evaluations use the US signal and transform it into the calculated attenuation indices.

As a key elastography-derived method, the controlled attenuation parameter (CAP) is used in modern clinical practice, which can be obtained simultaneously with an LSM by TE (Figure 2). Advantages of CAP include that it is a rapid, point-of-care assessment with moderate sensitivity and specificity for diagnosing fatty liver [5]. Because of its relatively low cost compared to MRI-PDFF, CAP is pontentialy suitable for MAFLD screening. The program provides numeric values expressed as dB/m ranging from 100 to 400 dB/m [5,19]. Two probes, M and XL, are available, and the automatic probe selection software included in the device recommends using the XL probe when the skin to liver capsule distance is >25 mm [5,19]. CAP is displayed only when the LSM is valid because it is computed from the ultrasound signals used for acquiring LSM [16].

Regarding the reliability criteria, the CAP-IQR should be <40 dB/m [4,7]. A well-designed study with MRI-PDFF comparison indicates that the use of CAP-IQR of <30 dB/m is even better [20,21]. More stringent reliability criteria such as CAP-IQR < 20 dB/m or CAP-IQR/M < 0.1 yielded the highest AUROC values with the highest confidence in the correct diagnosis of any steatosis. However, applying these criteria in clinical routine would have left a considerable proportion of patients (60–80%) with unreliable measurements [21].

In the reported studies, considerable variation is observed with respect to the proposed cut-off values for the assessment of liver steatosis grades. We mention some of the more important ones.

A study by Cai et al. determined that the average cut-off values for identifying patients with steatosis grades ≥S1, ≥S2, and S3 were 272 dB/m, 292 dB/m, and 308 dB/m [12]. Another study set the optimal CAP threshold for the detection of any liver steatosis as defined by MRI-PDFF ≥5% to 288 dB/m with a up to 90% diagnostic accuracy [20]. This was also confirmed by the study from Cai et al., as a median CAP of >279 dB/m was highly specific for the S1 presence. A large cohort study using liver biopsy as a reference observed a moderate performance for diagnosing S1 at an optimal cut-off of >246 dB/m [5]. The diagnostic performance of CAP for more severe hepatic steatosis (i.e., S2 and S3) tended to be worse and in line with similar reports [5,12,21].

A meta-analysis of data from more than 2000 patients showed that the optimal cut-off for distinguishing S0 from S1–3 is 246 dB/m with a moderate AUROC when using the 5% definition for S0 compared to an optimum of 249 dB/m when using the 10% definition. Cut-offs of 248 dB/m, 268 dB/m, and 280 dB/m for >S0, >S1, and >S2 were reported as proposed cut-off values [17]. If we want to achieve high sensitivity and specificity at the same time and identificate moderate steatosis (<S2), a cut-off of 331 dB/m is sufficient [13].

The literature is quite inconsistent when referring to the impact and interpretation of the factors involved in calculating the final CAP result. Probe selection impacts CAP values, and optimal tresholds for the fatty liver diagnosis are reported to be lower using the M probe vs. the XL probe [5,7,20]. There is still a significant lack of data regarding the limit values when applying the use of the XL probe [16]. In patients with a higher body mass (BMI) index and fibrosis stages, CAP tends to overestimate the grade of steatosis [4,12]. NASH patients have an increase in CAP values of 16 dB/m and diabetics an increase of 13 dB/m. CAP values are increased by 3.9 dB/m per BMI unit. Consequently, the group of authors recommends that the resulting deduces 10 dB/m from the CAP value for NASH patients, 10 dB/m for diabetes patients and adds/deduces 4.4 dB/m for each unit of BMI >/<25 kg/m^2^ over the range of 20–30 kg/m^2^ [22].

Two large meta-analyses were performed by Karlas et al. and Petroff et al. Karlas et al. report that the etiology, fibrosis stage, diabetes, and CAP-IQR were not found to be associated with discrepancies of the final CAP result. There were 15% of such patients in the meta-analysis [22]. Quite the opposite is reported by a recent large meta-analysis by Petroff et al. where they report that CAP values were independently affected by aetiology, diabetes, BMI, aspartate aminotransferase level, and gender. They state that CAP cut-offs cannot grade steatosis in patients with MAFLD adequately. They recommend that its value in a MAFLD screening setting should be studied, ideally with methods beyond the histological reference standard. They also state an interesting detail that in 90% of cases when the apparatus suggested the use of an XL probe, the patient had a diagnosis of MAFLD present [23].

Several attenuation imaging methods, such as attenuation imaging (ATI; Canon Medical Systems), ultrasound-guided attenuation parameter (UGAP; GE Healthcare Japan Co.), and attenuation coefficient (ATT; Hitachi), have been developed as new ultrasound-based methods for the assessment of liver steatosis. The pilot study has already examined ATI, showing moderate ability of detecting the presence and distinguishing different degrees of liver steatosis [24].

The field of liver steatosis assessment using the methods of elastography derived parameters is relatively new and rapidly evolving. The proposed cut-off values have not been study validated to the extent as to allow a clinical pathway proposition parallel to those of liver fibrosis and portal hypertension field. The above mentioned options for assessing the liver steatosis degree should therefore be used with caution in day-to-day clinical decisions until reliable cut-off values based on larger meta-analyzes are available.

## 5. Elastographic Assessment of Clinically Important Portal Hypertension

Elastography is crucial for the non-invasive detection of clinically important portal hypertension (CSPH) which represents a key complication and a milestone in the transition between clinical stages in the course of cACLD. Confirmation of CSPH can be performed by invasive measurement of hepatic venous pressure gradient (HVPG), or by confirming the presence of varices needing treatment (VNT) via upper endoscopy. VNT are defined as large (size ≥ 5 mm) or small varices (size < 5 mm) in case of Child-C cirrhosis, or if red spot signs are present [25].

Elastography allows a non-invasive assessment of the VNT occurrence likelihood. In doing so, we must be aware of important investigatory limitations, as “linear correlation between LSM and CSPH decreases for HVPG values higher than 12 mmHg and as such in advanced stages of cirrhosis, portal pressure becomes less dependent on intrahepatic resistance to portal flow due to progression of fibrosis, and more related to extra-hepatic factors, such as hyperdynamic circulation and splanchnic vasodilatation” [26].

In the field of non-invasive CSPH assessment, the recommendations of the Baveno expert group are used as a scientific basis. According to the Baveno VI consensus, CSPH may be suspected in patients with an LSM > 20–25 kPa, whereas the combination of an LSM < 20 kPa and a platelet count of > 150 × 10^9^ counts/mL may be used to safely rule out VNT. The risk of missing VNT when using these criteria is around 2%, and around 20% of endoscopies can be spared. The Baveno criteria were lately extended to propose tresholds of LSM < 25 kPa and platelet counts > 110 × 10^9^ counts/mL In this case, only 1.6% of cases with VNT are not recognized and 40% of gastroscopies can be spared. Later, the proposed NAFLD-cirrhosis criteria defined the separate LSM and platelets cut-off values for the M and XL probe. The spleen stiffness measurement (SSM) has been demonstrated as an excellent predictor of VNT, with a cut-off value of ≤46 kPa that reliably excludes VNT. Combined with the Baveno VI criteria, the algorithm was able to spare endoscopy in 44% of patients with <5% risk of missing VNT [4]. SSM may have the possibility to capture portal hypertension due to pre-sinusoidal or pre-hepatic causes usually not detected by LSM [25].

The following is an overview of some important field studies in recent years.

The study by Colecchia et al. performed a study analysis of the SSV and Baveno VI consensus combination and confirmed that 40% of upper gastrointestinal endoscopies in the observed cohort could be omitted. The inability to measure spleen stiffness with TE was up to 20%, which is why the investigation was performed US guided. Newer TE systems have eliminated this problem with the possibility of determining the spleen location by the concomitant US B-module picture (embedded US guiding system) [27]. The following multicenter study showed that <30 kPa for M probe and <25 kPa for XL probe were the best LSM thresholds while sustaining >110 × 10^9^ counts/mL as the best threshold for platelets and that the NAFLD cirrhosis criteria performed better than Baveno VI and expanded Baveno VI, enabling 68% of endoscopies to be spared while maintaining a similar rate of missed VNT [28]. The next study group proposes the calculation of the PLER score as an important contribution to the field. When platelets value is 17 times higher than the LSM value, VNT prevalence is less than 5% and endoscopy can be avoided. However, the reported missed VNT rate was borderline in the MAFLD group (7.4%). When PLER was less than 6.2, the VNT prevalence was greater than 5% in all clinical settings and endoscopy was mandatory. The gray zone between the referred PLER scores required a PLEASE calculation which includes age, sex, INR, and etiology of liver disease [29]. The Asian expert group demonstrated in a large study cohort of patients with viral-induced cACLD that LSM/SSM (LSSM) guided VNT screening proved to be non-inferior to universal endoscopic screening in patients with cirrhosis. An SSM cut-off value of <41.3 kPa was highly sensitive in ruling out VNT. Combining tresholds of LSM > 27.3 kPa and SSM > 40.8 kPa was found sensitive and specific to predict VNT. This combined strategy saved nearly 50% of patients from upper endoscopy [30]. Patients from the same cohort were then followed. The findings of the consecutive study suggest that patients defined as having a low risk of VNT by LSSM would have minimal future risk of incidental variceal bleeding (<1% in 3 years). Similarly, low risk of bleeding was observed in the low-risk group defined by Baveno VI criteria (1.7% in 3 years) [31]. The Austrian group evaluated the study by Kim et al. They reported that ΔSSM variations were related to HVPG variations after non-selective beta blockers (NSBB) therapy and that an SSM reduction of ≥10% was able to identify responders to NSBB therapy [26].

Most recently, patients with portosystemic shunting have been found to have an increased prevalence of esophageal varices, and that varices might be missed by transient elastography in those cases. Patients with preserved liver function (MELD 6–9 or Child-Pugh Stage A) and portosystemic shunting showed higher HVPG values and were found with significantly more portal hypertension related complications such as bleeding or ascites than those without shunting [25]. Much attention is also paid to the higher VNT prevalence in non-obese compared to obese patients. Baveno criteria had a worse performance in non-obese patients where the rate of missed VNT was of about 10%. Careful interpretation of non-invasive scores for ruling out VNT in non-obese patients with MAFLD-related cirrhosis is recommended [28]. In the unclear case, elastography and platelet count may be repeated after 6 months and reassessment of the condition should be performed [28,29].

Once again, a simplified flowchart is proposed as a summation of the topic and as proposition for the everyday clinical use (Figure 4).

## 6. Elastography and Other MAFLD Areas

Higher baseline LSM and ΔLSM were demonstrated to be associated with liver-related and all-cause mortality in MAFLD patients. Patients with LSM > 10 kPa were also found to be associated with higher occurrence of hepatic decompensation and HCC. Over time, the risk of decompensation increased in a dose-dependent fashion based on ΔLSM, ranging from 3.8% in those with a >20% ΔLSM decrease to 14.4% in those with a >20% ΔLSM increase [4,5,32,33]. Patients with LSM > 20 kPa have significantly diminished survival figures [3]. Significant increase in the risk of cardiovascular events has been reported in patients with higher grades of liver steatosis [4]. Reports in this area are somehow contradictory as they were not proven in a related publication (no influence of CAP measurements on the short-term liver-related events, cancer, or cardiovascular events prediction) [33]. CAP might be useful as a serial measurement in response to lifestyle or pharmacological/surgical interventions [13].

Although little attention has been paid to this area so far, steatosis may also be more common in patients, with cACLD being observed in 77% of the patients in the reported study [34]. However, a significant liver fat content was almost exclusively observed in patients with metabolic syndrome. The study showed that CAP can identify any grade of steatosis and S2–S3 steatosis with moderate accuracy, with results that did not differ from those obtained in patients with less severe chronic liver disease. The obtained data support the cut-off of 268 dB/m as accurate to rule out high-grade steatosis in cACLD patients [34,35]. These data could be particularly important in the light of prior publication by the same study group where they demonstrated that CAP can be used as a non-invasive predictor of prognosis in patients with cACLD and could improve the risk stratification for clinically relevant events and clinical decompensation provided by LSM. In the group where patients with cACLD of very different etiologies were included, a CAP value > 220 dB/m was significantly associated with decompensation events, even after subanalysis in the group of patients with LSM > 21 kPa was performed [36].

## 7. Combining Non-Invasive Tests

To improve the diagnostic accuracy of liver biopsy, recent approaches have led to combination tests as combination of two unrelated non-invasive tests can provide better accuracy and overcome limitations of a single test [6]. One strategy is to combine blood test results with ultrasound elastography and imaging; a notable example includes FAST score: combination of serum aspartate aminotransferase (AST) with LSM and CAP value. An alternate strategy is to use these tests in a tandem approach, starting with a readily available blood-based test followed by elastography. A three-step strategy by performing a second elastography technique in patients with unreliable or gray zone results further reduced the need for liver biopsy [37]. With a three-step strategy in which FIB-4 is combined with two elastography techniques, a study analysis on a large cohort of MAFLD patients proved that the percentage of unclassified patients is exceptionally low (<5%), with a diagnostic accuracy greater than 80% [8]. When TE and serum biomarker results are in concordance, the diagnostic accuracy for significant fibrosis is reportedly increased, but not for cirrhosis. In cases of unexplained discordance, a liver biopsy should be performed if the results would change patient management [6].

## 8. Upcoming Trends in the Field of Ultrasound Elastography

Just recently, TE (Fibroscan, Echosens, Paris, France) has launched Smart Exam equipment, which should improve reliability in the diagnosis and monitoring of steatosis with Continuous CAP option. This option is also meant to extend TE usage among severely obese patients because of its ability of continuous measurement of CAP during the entire examination. In addition, when CAP measurement doesn’t meet the quality criteria they are automatically rejected. This method might also increase the ability to assess liver fibrosis and steatosis as it is able to measure both parameters at 28% greater depth than the previous models. Study reports mention the possibility of using artificial intelligence (CNN) programs for ultrasound image analysis. An interesting study compared ultrasound images of severely obese patients collected before bariatric surgery. A biopsy was taken from each patient and the presence of liver steatosis was pathologically assessed. A comparative CNN program then analyzed the ultrasound-obtained image and reported the presence or absence of liver steatosis with great reliability and accuracy [38]. A related study used the CNN program to assess the composition of the histological sample. Pathologists otherwise involved in large clinical studies performed sample analysis and labeled fields of steatosis, fibrosis, and inflammation. The CNN program was then able to process the histological sample with similar accuracy. The method would be useful for large MAFLD studies that require harmonized opinion of three pathologists regarding liver specimen composition [39]. 3D elastography methods are being experimented and are likely to receive transition to the clinical field [40]. Some study models evaluated the relationship between viscoelasticity and poroelasticity of liver tissue. Viscosity causes dispersion of the shear wave. Shear wave dispersion ultrasound vibrometry (SDUV) and a time harmonic elastography (THE) method were reported for measuring liver viscoelasticity in wide soft tissue windows and at greater depths [41]. Further research is needed to evaluate the diagnostic potential of liver viscosity, but preliminary evidence suggests that it may be helpful in detecting necroinflammation in the liver [41]. Dispersion slope (DS) is related to tissue viscosity and can be measured with a new version of 2D-SWE. DS had high diagnostic performance in discriminating the presence and degree of lobular inflammation in the reported study [40]. There are several reported study models regarding US attenuation imaging. The Acoustic Structure Quantification (ASQ) software is based on the concept that fat droplets within the liver differ from the normal liver parenchyma in terms of ultrasound echo amplitude distribution, a parameter calculated by using ASQ, namely focal disturbance ratio showed a strong correlation with liver fat fraction measured by MRI spectroscopy, and an excellent discriminative ability for steatosis [41]. Similar imaging model reports that the BSC index measures the returned ultrasound energy from tissue and provides a quantitative parameter analogous to echogenicity of the liver tissue [3]. The tissue scatter-distribution imaging (TSI-p) and tissue attenuation imaging parameter (TAI-p) are reported to reflect the local concentration and arrangement of US signal scattering when passing through the liver tissue infiltrated by fat droplets. In a study by Jeon et al. where CAP measurements were used as a reference with mostly HBV patients included, both indices were well correlated when diagnosing the presence and different grades of steatosis. The authors mention the possibility of the effect of liver fibrosis on the final US attenuation result [42].

## 9. Conclusions

Elastography is clinically useful, point of care, and affordable non-invasive method for mass and rapid assessment of liver fibrosis and steatosis presence/degree, and the presence of CSPH/VNT. The method shows excellent results when transferring established data to the MAFLD cohort, with its negative prognostic value being the most importan. The terminological and diagnostic change from NAFLD to MAFLD will require a detailed analysis regarding the predictive thresholds value already defined in NAFLD study cohorts. Due to significant limitations of the results’ positive predictive value, the inability of elastography to assess the presence and degree of NASH activity, and the arrival of many different (poorly tested) models on the market, we must also always evaluate many aspects of the proposed clinical methods and make any clinical decision regarding further patients’ management with most care and precision. The applicability of the field is expected to expand to the field of national prevention programs which could select a subgroup of MAFLD patients that need further evaluation. Numerous upcoming studies and consecutive meta-analyses in this field will play an important role and are certain to bring answers to some of the dilemmas that were briefly presented in the article.

## Figures and Tables

**Figure 1 medicina-57-00516-f001:**
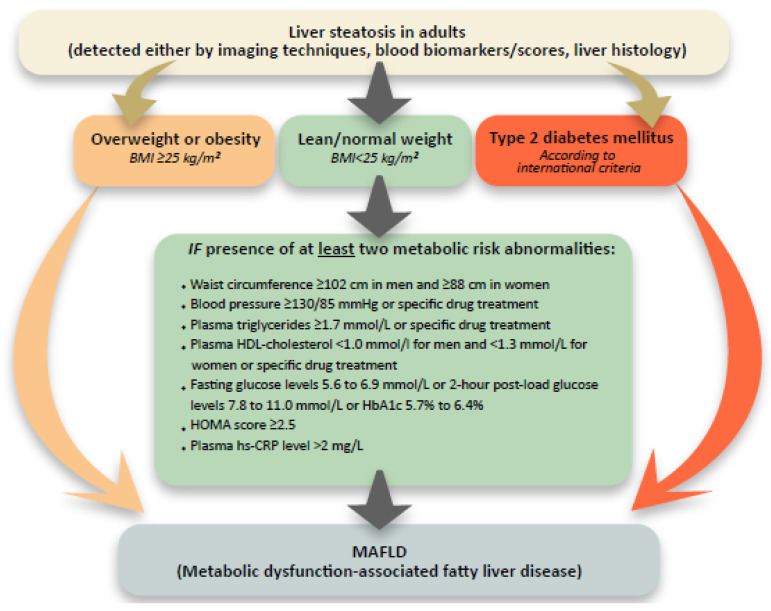
MAFLD criteria. Addapted from Eslam, M., et al. A new definition for metabolic dysfunction-associated fatty liver disease: An international expert consensus statement [1]. BMI—body mass index. HOMA score—Homeostatic Model Assessment for Insulin Resistance score. hsCRP—high-sensitivity C-reactive protein.

**Figure 2 medicina-57-00516-f002:**
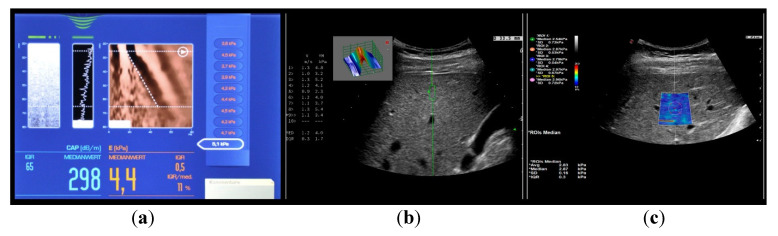
US based elastographic techniques. (**a**) transient elastography; (**b**) point shear wave elastography; (**c**) 2D shear wave elastography.

**Figure 3 medicina-57-00516-f003:**
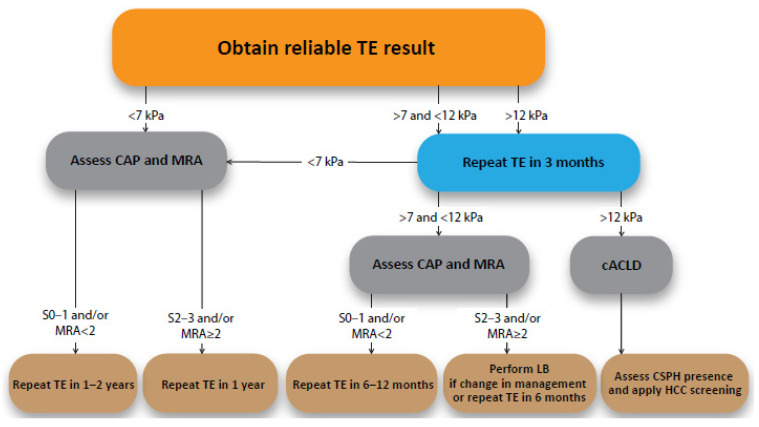
Assessing liver fibrosis in MAFLD patient. Any reliable TE >21 kPa is sufficient to rule cACLD in. TE—transient elastography. CAP—controlled attenuation parameter. MRA—metabolic risk abnormalities (see Figure 1). cACLD—compensated advance chronic liver disease. LB—liver biopsy. CSPH—clinicaly significant portal hypertension. HCC—hepatocelular carcinoma.

**Figure 4 medicina-57-00516-f004:**
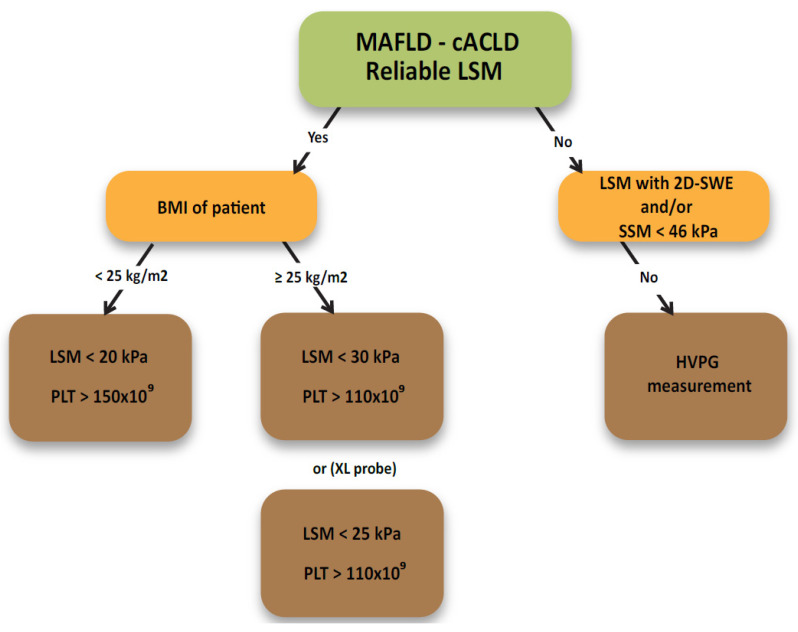
Elastographic evaluation of VNT presence. Sequential SSM-VNT evaluation is possible after every LSM-VNT evaluation protocol. MAFLD—metabolic dysfunction-associated fatty liver disease. cACLD—compensated advanced chronic liver disease. LSM—liver stiffness measurement. SSM—spleen stiffness measurement. BMI—body mass index. PLT—platelets count (counts per microliter of blood). HVPG—hepatic venous pressure gradient.

## Data Availability

Not applicable.

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
