# Peer review of "Ultrasound Elastography—Cornerstone of Non-Invasive Metabolic Dysfunction-Associated Fatty Liver Disease Assessment"

_medicina, 2021, doi:10.3390/medicina57060516_

Round 1

Reviewer 1 Report

The author has written a position paper where the topic is liver elastography and attenuation parameters in Metabolic syndrome and a the abbreviation Metabolic Dysfunction-Associated Liver Disease (MAFLD) is used consequently instead of the more known and used Non-Alcoholic Fatty Liver Disease, NAFLD, which may be a misnomer describing an exclusion diagnosis and may contain several different entities. This "political" statement may have a relevance. The author propose algorithms for follow-up in the  form of flow-charts. MAFLD is probably the liver disease with the  highest rising prevalenvce  in the world, and knowledge about how US elastography can be used to monitor its development and status is very useful and valuable, as it may help predict the prognosis, and help patients change lifestyle and medical doctors in the  follow-up. 

The authors seem to lean heavily on TE (Fibroscan), and as a reader I suppose they have first hand knowledge with this method, and maybe not with the other, more image based methods. It is true that TE has good documentation, but there is more available for pSWE and 2D SWE than authors have referred to.

Major comments:

1. In the  first part, the most used different elaststography methods are summarized, but there is given no references to normal liver stiffness values or to studies that have compared the methods head-to-head. This would improve the second part: Ultrasonography and MAFLD evaluation. Please consider references on this such as: Liver elasticity in healthy individuals by two novel shear-wave elastography systems-Comparison by age, gender, BMI and number of measurements.

Mulabecirovic A, Mjelle AB, Gilja OH, Vesterhus M, Havre RF.PLoS One. 2018 Sep 14;13(9):e0203486. doi: 10.1371/journal.pone.0203486. eCollection 2018.PMID: 30216377 and Normal Liver Stiffness Values in Children: A Comparison of Three Different Elastography Methods. Mjelle AB, Mulabecirovic A, Havre RF, Rosendahl K, Juliusson PB, Olafsdottir E, Gilja OH, Vesterhus M.J Pediatr Gastroenterol Nutr. 2019 May;68(5):706-712. doi: 10.1097/MPG.0000000000002320.PMID: 30889132   2. P. 4 L115-117 "However, a study on phantoms recently showed  that there is a significant difference in LSM estimates among systems, but no statistically significant differences were found among observers using the same system, and they also reported very good agreement between systems [11]. "

It has been described that especially in TE, the measurements are much higher in kP for harder tisses than for pWE and 2D SWE. The variability may also  be very  high, particularly  for LSM values > 10.  You  describe several influencing factors and sources of measurement variability  and this is also the case, the main problem with liver biopsy is the invasive nature of it and the risk of sampling error. Perhaps this reference would be of interest too: In Vitro Comparison of Five Different Elastography Systems for Clinical Applications, Using Strain and Shear Wave Technology. Mulabecirovic A, Vesterhus M, Gilja OH, Havre RF.Ultrasound Med Biol. 2016 Nov;42(11):2572-2588. doi: 10.1016/j.ultrasmedbio.2016.07.002. Epub 2016 Aug 25.PMID: 27570209

3. P. 5. L 162: ..."significant fibrosis (≥F3)" Why has the  author selected >=F3 as significant fibrosis? For other entities such as Hepatitis C induced liver fibrosis, Significant fibrosis is used from >=F2 (Metavir score)? (e.g. by  Ferraioli et al.)

4. In section 4. Elastographic assessment of liver steatosis. P. 6, L 213-215: "As a key elastographic method, the controlled attenuation parameter (CAP) is used in modern clinical practice, which can be obtained simultaneously with an LSM by TE (Figure 2). Advantages of CAP include that it is a rapid, point-of-care assessment with moderate sensitivity and specificity for diagnosing fatty liver [5]."  CAP and all other attenuation measurement along the US beams are not elastography based methods, they are only calculating attenuation differences with  depth, so both the sub-headline and the paragraph above should be changed, even if Controlled Attenuation Parameter (CAP) is available on the Fibroscan, it is calculated by  the A- mode attenuation of US and has nothing to  do  with LSM or elastography. The parameter has long been available in regular US scanners known e.g. on GE scanners as Echo-levels. (Quantitative measurement of ultrasound attenuation and Hepato-Renal Index in Non-Alcoholic Fatty Liver Disease.

von Volkmann HL, Havre RF, Løberg EM, Haaland T, Immervoll H, Haukeland JW, Hausken T, Gilja OH.Med Ultrason. 2013 Mar;15(1):16-22. doi: 10.11152/mu.2013.2066.151.hlv1qmu2.PMID: 23486619)

It is important that attenuation as in fattly liver makes accurate and repeatable LSM measurements more difficult to obtain. 

5. Legend to fig. 3 ( flow-chart) seems very unfinished and does not explain the figure well. It is unusual and unnecessary to repeat references in figure legends.(" Figure 3. Assessing liver fibrosis in MAFLD patient. Note the important disadvantage of the chart─evaluation of the 189 inflammation (NASH) presence and the inflammatory activity degree. Inclusion of proposed formulas reported by Papa Theodoridi et al. [14] is recommended. Any reliable TE >21 kPa is sufficient to rule cACLD in. ")?

Minor comments: 

  1. Conclusion: Please also separate between elasticity  measurements (which are share -wave speed derived in all the  methods described, also the TE) and attenuation indexes which are indicating level of steatosis. This is not the  same method, but steatosis may affect both. The statement of combining blood samples, elastography and attenuation indexes, (no matter who the  producer of the  equipment is) for the  benefit of diagnosing and follow-up of patients with metabolic syndrome is important and could improve this in practice
  2. P12. L443: Please rephrase, weak syntax. "...with its negative prognostic value (exclusion) being of the most importance. 
  3. P12, L 453-454: Please rephrase last sentence as it has a weak syntax and is just a phrase: "....as we are certain to expect some further answers regarding dilemmas that were briefly presented in the article." 

Author Response

Dear Reviewer and Editors,

thank you for your thoughtful comments which enable us to improve the quality of the proposed article.

Below are the point by point responses and proposed corrections regarding the Review 1.

The author has written a position paper where the topic is liver elastography and attenuation parameters in Metabolic syndrome and a the abbreviation Metabolic Dysfunction-Associated Liver Disease (MAFLD) is used consequently instead of the more known and used Non-Alcoholic Fatty Liver Disease, NAFLD, which may be a misnomer describing an exclusion diagnosis and may contain several different entities. This "political" statement may have a relevance. The author propose algorithms for follow-up in the  form of flow-charts. MAFLD is probably the liver disease with the  highest rising prevalenvce  in the world, and knowledge about how US elastography can be used to monitor its development and status is very useful and valuable, as it may help predict the prognosis, and help patients change lifestyle and medical doctors in the  follow-up. 

Response: Thank you for a fine presented and short cover of the article. Our only thought would be (knowing that indeed it is a "political" issue in the field of hepatology) that it was meant to be more of a – following a trend and the sense of the hepatology future – instead.

The authors seem to lean heavily on TE (Fibroscan), and as a reader I suppose they have first hand knowledge with this method, and maybe not with the other, more image based methods. It is true that TE has good documentation, but there is more available for pSWE and 2D SWE than authors have referred to.

Response: I understand that the topic could misslead the reader in this direction, but no. In quite oposite – our center relies on pSWE and 2D-SWE US techniques while surely combining it with the TE apparatus. The TE method is more heavily menitoned only because it is the most studied and validated method of the field.

Major comments:

  1. In the  first part, the most used different elaststography methods are summarized, but there is given no references to normal liver stiffness values or to studies that have compared the methods head-to-head. This would improve the second part: Ultrasonography and MAFLD evaluation. Please consider references on this such as: Liver elasticity in healthy individuals by two novel shear-wave elastography systems-Comparison by age, gender, BMI and number of measurements.

Mulabecirovic A, Mjelle AB, Gilja OH, Vesterhus M, Havre RF.PLoS One. 2018 Sep 14;13(9):e0203486. doi: 10.1371/journal.pone.0203486. eCollection 2018.PMID: 30216377 and Normal Liver Stiffness Values in Children: A Comparison of Three Different Elastography Methods. Mjelle AB, Mulabecirovic A, Havre RF, Rosendahl K, Juliusson PB, Olafsdottir E, Gilja OH, Vesterhus M.J Pediatr Gastroenterol Nutr. 2019 May;68(5):706-712. doi: 10.1097/MPG.0000000000002320.PMID: 30889132   2. P. 4 L115-117 "However, a study on phantoms recently showed  that there is a significant difference in LSM estimates among systems, but no statistically significant differences were found among observers using the same system, and they also reported very good agreement between systems [11]. "

It has been described that especially in TE, the measurements are much higher in kP for harder tisses than for pWE and 2D SWE. The variability may also  be very  high, particularly  for LSM values > 10.  You  describe several influencing factors and sources of measurement variability  and this is also the case, the main problem with liver biopsy is the invasive nature of it and the risk of sampling error. Perhaps this reference would be of interest too: In Vitro Comparison of Five Different Elastography Systems for Clinical Applications, Using Strain and Shear Wave Technology. Mulabecirovic A, Vesterhus M, Gilja OH, Havre RF.Ultrasound Med Biol. 2016 Nov;42(11):2572-2588. doi: 10.1016/j.ultrasmedbio.2016.07.002. Epub 2016 Aug 25.PMID: 27570209

Response: This a valuable comment. We added the text in the article to better support the idea mentioned by the reviewer. We are all well aware that the three (if leaving the strain and apri aside, because they have much lesser impact and minor position in the adult elasto world) of the proposed tecnniques do no yield the same results and that the biopsy validated studies will probably differ the cut off values between the three of them in the future. Yet, to make the article more of a clinicaly oriented one, I would suggest to stick to the original idea of the article since when explaining all the (important!) differences, the article could easly miss the focus of its basic idea.

Also, the proposed two articles were surely read by the author. On one side, as I mentioned to the other reviewer, ours is the field of the adult, not pediatric hepatology population. Since the article was solicitated for the special issue of Medicina, we received clear instructions regarding the theme of the article. So, when deciding which articles to include and which to exclude at the end, the beformentioned two (the reviewers study group, I suppose?) were not included. If it is of importance, we can include them in the article for sure.

  1. P. 5. L 162: ..."significant fibrosis (≥F3)" Why has the  author selected >=F3 as significant fibrosis? For other entities such as Hepatitis C induced liver fibrosis, Significant fibrosis is used from >=F2 (Metavir score)? (e.g. by  Ferraioli et al.)

Response: we corrected the typo (P5 L162). Thank you for noticing.

  1. In section 4. Elastographic assessment of liver steatosis. P. 6, L 213-215: "As a key elastographic method, the controlled attenuation parameter (CAP) is used in modern clinical practice, which can be obtained simultaneously with an LSM by TE (Figure 2). Advantages of CAP include that it is a rapid, point-of-care assessment with moderate sensitivity and specificity for diagnosing fatty liver [5]."  CAP and all other attenuation measurement along the US beams are not elastography based methods, they are only calculating attenuation differences with  depth, so both the sub-headline and the paragraph above should be changed, even if Controlled Attenuation Parameter (CAP) is available on the Fibroscan, it is calculated by  the A- mode attenuation of US and has nothing to  do  with LSM or elastography. The parameter has long been available in regular US scanners known e.g. on GE scanners as Echo-levels. (Quantitative measurement of ultrasound attenuation and Hepato-Renal Index in Non-Alcoholic Fatty Liver Disease.

von Volkmann HL, Havre RF, Løberg EM, Haaland T, Immervoll H, Haukeland JW, Hausken T, Gilja OH.Med Ultrason. 2013 Mar;15(1):16-22. doi: 10.11152/mu.2013.2066.151.hlv1qmu2.PMID: 23486619)

It is important that attenuation as in fattly liver makes accurate and repeatable LSM measurements more difficult to obtain. 

Response: I think that the mentioned topic is a relevant one. Yet, even though we are all well aware that strictly speaking CAP is not elasto technique, it is still calculated from the elasto based method. Just like LSM. So, like all the basic and important studies – why not call it a elasto derived technique? I added a simple explanation in the article, to not misslead any of the readers. Regarding the second part of the comment – the US based techniques of the steatosis evaluation (similar to the two mentioned) are covered in the article, but just some of the novel calculation techniques are cited. The topic of fatty liver – difficult to measure LSM – is also covered in the article. Again, the article from von Volkmann is a very fine presented one and was of course read by the authors when writing the article.

  1. Legend to fig. 3 ( flow-chart) seems very unfinished and does not explain the figure well. It is unusual and unnecessary to repeat references in figure legends.(" Figure 3. Assessing liver fibrosis in MAFLD patient. Note the important disadvantage of the chart─evaluation of the 189 inflammation (NASH) presence and the inflammatory activity degree. Inclusion of proposed formulas reported by Papa Theodoridi et al. [14] is recommended. Any reliable TE >21 kPa is sufficient to rule cACLD in. ")?

Response: We agree that referencing in the figure legends is not common, yet it is not rare either. I would ask for a more clear instrucion of what may be unfinished and not well explained in the legend? Should we just leave Assessing liver fibrosis in MAFLD patient. Any reliable TE >21 kPa is sufficient to rule cACLD in. ") and put the middle sentence to the text instead under the Figure (proposed in the rewised uploaded version - see the corrected Figure)?

Minor comments: 

  1. Conclusion: Please also separate between elasticity  measurements (which are share -wave speed derived in all the  methods described, also the TE) and attenuation indexes which are indicating level of steatosis. This is not the  same method, but steatosis may affect both. The statement of combining blood samples, elastography and attenuation indexes, (no matter who the  producer of the  equipment is) for the  benefit of diagnosing and follow-up of patients with metabolic syndrome is important and could improve this in practice

Reponse: we did some minor corrections in the text. Hopefully, this would improve the understanding of the mentioned issue.

  1. P12. L443: Please rephrase, weak syntax. "...with its negative prognostic value (exclusion) being of the most importance. 

Response: it is difficult to understand what may be read as a weak syntax to the reviewer, but we tried to improve the sentence.

  1. P12, L 453-454: Please rephrase last sentence as it has a weak syntax and is just a phrase: "....as we are certain to expect some further answers regarding dilemmas that were briefly presented in the article." 

Response: the answer would be similar to the previous response.

Andrej Hari

Reviewer 2 Report

The manuscript entitled “Elastography – cornerstone of non-invasive metabolic 2 dysfunction-associated fatty liver disease assessment” submitted by Hari A summarized the recent advancement in the noninvasive technology used for the diagnosis of metabolic dysfunction-associated fatty liver disease (MAFLD). The manuscript is well planned and written and the author have cited the appropriate and recent literature to support his views. However, the author mainly focused on the diagnosis of adult NAFLD/MAFLD and completely ignored to mention about pediatric NAFLD which have a unique histopathology. Hence I highly recommend author to summarize few of the recently published papers [PMID: 31278377, PMID: 26031832, PMID: 28472948 and PMID: 33081177] in this area.  

Author Response

Dear Reviewer and Editors,

thank you for your thoughtful comments which enable us to improve the quality of the written article.

I would gladly addapt the article as suggested (cover the pediatric part of the proposed topic) but since the article is solicitated for the special thematic issue of Medicina and I was instructed to write of the adult population diagnostics only (rightly so - I am not a pediatrician trained in hepatology) I can not provide the recommended topic in my article.

With kind regards,

Andrej Hari

Reviewer 3 Report

Journal: Medicina

Article: Elastography – cornerstone of non-invasive metabolic dysfunction-associated fatty liver disease assessment.

Overall Impression:

The authors have written a review article on the use of ultrasound elastography for the assessment of metabolic dysfunction-associated fatty liver disease. They have provided useful flow-charts for assessing MAFLD-cACLD, obtaining reliable transient elastography results, and measuring liver steatosis in adults. However, the report seems to primarily be reporting results and does not have sufficient summary statements and concluding sentences to help guide readers. Furthermore, it lacks details in the technical aspects of elastography and lacks sufficient citations to support some of the statements in the text. My specific comments are listed below. In addition to these I would recommend this paper be proofread by a native English speaker to fix some of the terminology used.

Specific comments:

  1. Provide country in affiliation (Slovenia).
  2. Abstract (Ln 8): What is meant by procedure types here? Do the authors mean different methods of elastography or different surgical procedures?
  3. Pg. 1. Ln. 21. Citation needed for this statement.
  4. Pg. 1. Ln. 28. Please give the name of the group of experts in the text.
  5. Pg. 1. Ln. 32. Unclear what is meant by “In a sense of an article better recognition”.
  6. Figure 1. Make sure units are correct (6.9 mol/L) should be mmol/L.
  7. The use of the word “milestone” is not conventional here. Consider revising.
  8. Pg. 2. Ln. 44. “SAF” is an undefined acronym.
  9. Pg 2. Ln. 54. Unclear what “Fore” is and a citation needs to be included.
  10. Why were MRI and CT methods not included in this review? Especially MR elastography, which is considered by some to be the gold standard for this application? Consider revising or providing an explanation for this.
  11. The authors should consider including the terminology “Ultrasound Elastography” in their title to make it more clear to readers.
  12. Pg. 2. Ln. 56. Elastography is not a group.
  13. Overall a better explanation of the similarities and differences of transient elastography, point shear wave elastography, and 2D shear wave elastography need to be described. Also, this section was missing citations for each of these techniques. What do each of these measure?
  14. In general throughout the text verbiage such as “good”, “excellent” etc. should not be used unless they are defined. Ideally these would be exchanged for reported exact values of the quantities that were measured.
  15. Pg. 3. Ln. 89-90. Citations needed for claims of unreliable and reliable measurements.
  16. Pg. 3. Ln. 85-118. Consider summarizing these results in a table.
  17. Pg. 4. Ln. 100. Citation needed for scan failure rate.
  18. Ln. 102. Citation needed for 100 examinations study.
  19. Ln. 104. Citation needed for MAFLD study.
  20. Ln. 123. Citation needed for this statement.
  21. Ln. 196-203. Consider summarizing this section in a table.
  22. Ln. 209. Provide the exact methods and the justification for making this claim.
  23. Ln. 236. Report exact values of accuracy and how that was accessed.
  24. Units are missing in platelet counts in text and figure. I am assuming this is counts/mL?
  25. Ln. 398. Smart Exam seems like jargon that is used by manufacturers. What does this mean? What is the technology behind it? This needs to be explained and citations need to be included.
  26. Ln. 400. Type “because”.
  27. Upcoming trends: At times in this section it is unclear if elastography approaches are being discussed or just ultrasound advancements are being discussed. Make this clear.
  28. Ln. 420. Citation needed.
  29. Again, I would emphasize that more effort needs to be made to ensure each topic has a good introductory and concluding statement. I strongly suggest that new summary tables be included for each section to accommodate the existing figures.

Author Response

Dear Reviewer and Editors, thank you for your thoughtful comments that help us to improve the quality of our article. Provided below is the written response for your consideration. 1.  The authors have written a review article on the use of ultrasound elastography for the assessment of metabolic dysfunction-associated fatty liver disease. They have provided useful flow-charts for assessing MAFLD-cACLD, obtaining reliable transient elastography results, and measuring liver steatosis in adults. However, the report seems to primarily be reporting results and does not have sufficient summary statements and concluding sentences to help guide readers. Furthermore, it lacks details in the technical aspects of elastography and lacks sufficient citations to support some of the statements in the text. My specific comments are listed below. In addition to these I would recommend this paper be proofread by a native English speaker to fix some of the terminology used.   Response: this is a valuable comment as it states some of the weaker points of the article. As to the - sufficient summary statements and concluding sentences to help guide readers. Furthermore, it lacks details in the technical aspects of elastography - part, the aricle is revised and proposed in an improved version. Some of the details will be missing for sure as it is a solicitated article meant to be part of a themed Medicina and instructions were given to the authors as which parts to cover. Some of the comments proposed by the reviwer could be translated to the other articles in the themed issue of Medicina.  
Specific comments:
1. Provide country in affiliation (Slovenia). Response: we added the country in the affiliation part.   2. Abstract (Ln 8): What is meant by procedure types here? Do the authors mean different methods of elastography or different surgical procedures? Response: procedure types are meant as different metods of elastography for sure. The text is changed according to the comment.   3. Pg. 1. Ln. 21. Citation needed for this statement. Response: citation is provided for the statement.   4. Pg. 1. Ln. 28. Please give the name of the group of experts in the text. Response: Group of the experts is mentioned in the revised text.   5. Pg. 1. Ln. 32. Unclear what is meant by “In a sense of an article better recognition”. Response: It is meant as - to make an article more easily recognized when dealing with the NAFLD/MAFLD discussion.   6. Figure 1. Make sure units are correct (6.9 mol/L) should be mmol/L. Response: Units will be corrected in the finalised version of the picture.   7. The use of the word “milestone” is not conventional here. Response: a more conventional word is use in the revised version instead.   8. Pg. 2. Ln. 44. “SAF” is an undefined acronym. Response: Acronym SAF is explained in the revised version of the text.   9. Pg 2. Ln. 54. Unclear what “Fore” is and a citation needs to be included. Response: The fore: as in coming to the leading position.   10. Why were MRI and CT methods not included in this review? Especially MR elastography, which is considered by some to be the gold standard for this application? Consider revising or providing an explanation for this.   Response: As mentioned earlier, this is an invited paper covering the specific themed issue of the NAFLD/MAFLD area. Some other aricles might be covering the proposed topic.   11. The authors should consider including the terminology “Ultrasound Elastography” in their title to make it more clear to readers. Response: this is a valuable comment and we have changed the title accordingly.   12. Pg. 2. Ln. 56. Elastography is not a group. Response: structure of the sentence corrected in the revised version of the paper.   13. Overall a better explanation of the similarities and differences of transient elastography, point shear wave elastography, and 2D shear wave elastography need to be described. Also, this section was missing citations for each of these techniques. What do each of these measure? Response: there is a short text added to discuss this specific topic. Otherwise, as previously stated, this might be part of some other themed articles as the authors were instructed to provide an owerview of the entitled topic. The citations are stated at the end of the paragraphs. They measure the stiffness of the organ that is of operators interest.   14. In general throughout the text verbiage such as “good”, “excellent” etc. should not be used unless they are defined. Ideally these would be exchanged for reported exact values of the quantities that were measured. Response: Some of the phrasing is corrected in the revised version of the aricle.  
15. Pg. 3. Ln. 89-90. Citations needed for claims of unreliable and reliable measurements. Response: Citations are stated at the end of the paragraph as all the statements are part of the same literature.   16. Pg. 3. Ln. 85-118. Consider summarizing these results in a table. Response: once again, a valuable comment. The authors were discussing to add a similar table to the article, but decided against it because of the lenght of the existing text. It can be still added for sure if it is of utmost importance for the understanding of the article.   17. Pg. 4. Ln. 100. Citation needed for scan failure rate. Response: Citation is stated at the end of the paragraph as all the statements are part of the same literature.   18. Ln. 102. Citation needed for 100 examinations study. Response: a valid comment. We are all aware that no such study exists. It is a number proposed by the authors in the cited reference.   19. Ln. 104. Citation needed for MAFLD study. Response: Citation is stated at the end of the paragraph as all the statements are part of the same literature.   20. Ln. 123. Citation needed for this statement. Response: Citation is stated at the end of the paragraph as all the statements are part of the same literature.   21. Ln. 196-203. Consider summarizing this section in a table. Response: A vauable comment. Once again, our response would be simmilar to the one under the Nr 16.   22. Ln. 209. Provide the exact methods and the justification for making this claim. Response: the revised text is improved to provide a better understanding of the claim.   23. Ln. 236. Report exact values of accuracy and how that was accessed. Response: the revised text is improved to provide a better understanding of the claim.   24. Units are missing in platelet counts in text and figure. I am assuming this is counts/mL? Response: units are added in the text and will be added in the final version of the Figure.   25. Ln. 398. Smart Exam seems like jargon that is used by manufacturers. What does this mean? What is the technology behind it? This needs to be explained and citations need to be included. Response: correct. The revised text is improved to provide a better understanding of the claim.   26. Ln. 400. Type “because”. Response: the revised text was corrected.   27. Upcoming trends: At times in this section it is unclear if elastography approaches are being discussed or just ultrasound advancements are being discussed. Make this clear. Response: The revised text is improved to provide a better understanding of the claim   28. Ln. 420. Citation needed. Response: citation is provided in the revised text.   29. Again, I would emphasize that more effort needs to be made to ensure each topic has a good introductory and concluding statement. I strongly suggest that new summary tables be included for each section to accommodate the existing figures. Response: hopefully, some of the changes in the revised version of the article will provide the adequate potential to the reviewers comment. Regarding the tables, our response would be stated under the Nr 16.   Kind regards,   ---
Andrej Hari

Round 2

Reviewer 3 Report

I recommend that the authors attempt to respond to my original comments carefully and within the manuscript itself. 

In scientific writing citations go next to the claims made, not at the end of the paragraph like with other writing styles.

If there is no study describing that intraobserver agreement increases after 100 examinations, what is this statement based on?

Summary statements and tables need to be placed at the end of every paragraph. Figures need to be corrected. 

This manuscript needs extensive editing of English language and style.
